# Convolution Neural Network with Coordinate Attention for Real-Time Wound Segmentation and Automatic Wound Assessment

**DOI:** 10.3390/healthcare11091205

**Published:** 2023-04-23

**Authors:** Yi Sun, Wenzhong Lou, Wenlong Ma, Fei Zhao, Zilong Su

**Affiliations:** 1National Key Laboratory of Electro-Mechanics Engineering and Control, School of Mechatronical Engineering, Beijing Institute of Technology, Beijing 100010, China; 2Beijing Institute of Technology Chongqing Innovation Center, Chongqing 401120, China

**Keywords:** convolutional neural networks, coordinate attention, wound segmentation, area assessment

## Abstract

Background: Wound treatment in emergency care requires the rapid assessment of wound size by medical staff. Limited medical resources and the empirical assessment of wounds can delay the treatment of patients, and manual contact measurement methods are often inaccurate and susceptible to wound infection. This study aimed to prepare an Automatic Wound Segmentation Assessment (AWSA) framework for real-time wound segmentation and automatic wound region estimation. Methods: This method comprised a short-term dense concatenate classification network (STDC-Net) as the backbone, realizing a segmentation accuracy–prediction speed trade-off. A coordinated attention mechanism was introduced to further improve the network segmentation performance. A functional relationship model between prior graphics pixels and shooting heights was constructed to achieve wound area measurement. Finally, extensive experiments on two types of wound datasets were conducted. Results: The experimental results showed that real-time AWSA outperformed state-of-the-art methods such as mAP, mIoU, recall, and dice score. The AUC value, which reflected the comprehensive segmentation ability, also reached the highest level of about 99.5%. The FPS values of our proposed segmentation method in the two datasets were 100.08 and 102.11, respectively, which were about 42% higher than those of the second-ranked method, reflecting better real-time performance. Moreover, real-time AWSA could automatically estimate the wound area in square centimeters with a relative error of only about 3.1%. Conclusion: The real-time AWSA method used the STDC-Net classification network as its backbone and improved the network processing speed while accurately segmenting the wound, realizing a segmentation accuracy–prediction speed trade-off.

## 1. Introduction

Wound treatment in modern emergency environments, such as battlefields, fire disasters, and earthquakes, has distinct characteristics: more wounded but fewer medical personnel, fewer medical resources, and difficult medical evacuation. These factors make large-scale wound care extremely difficult in emergency situations [1]. Meanwhile, body surface wounds, such as scratches and burns, can lead to infection and poor blood circulation if not treated in time or amputation in severe cases [2,3,4]. Generally, the area and depth of the wound are manually estimated by medical staff [5]. Another method is to use a regular video camera to photograph the wound with a reference scale (such as a tape or a ruler) and upload it. Medical experts can judge the boundary and area of the wound and decide on its treatment. However, the manual segmentation of the wound area is complex and time-consuming, which delays the treatment of many wounded patients, and the contact wound measurement method is prone to wound infection. Therefore, a real-time and accurate tool is needed to assist in emergency medical care. The sequelae caused by the wound can be minimized by judging the state of the wound, transmitting wound information in time, and taking targeted treatment measures.

As technological advances in smartphones, computing storage devices, and clinical devices have improved the quality of image information [6,7], the computer-aided automatic segmentation and measurement of wound size have become new methods for accurate wound assessment. In particular, artificial intelligence technology has proved its efficiency and high performance in automatic image classification via machine learning methods. Artificial intelligence effectively removes a large amount of redundant information, analyzes and judges the state of the wound according to the wound image data, and assists telemedicine experts in preparing the best treatment plan so as not to miss the “golden 30 min” of emergency treatment. Deep learning (DL) is an extension of machine learning that mainly focuses on the automatic extraction and classification of image features and has achieved great success in many applications, especially in healthcare [8,9]. The introduction of DL techniques has motivated several researchers to use convolutional neural networks (CNNs) in the medical domain [10]. CNNs are a powerful tool for image processing owing to their good feature representation capability [11,12,13,14]. Photographic images have been used to recognize melanomas by segmenting [15,16,17] or classifying them; they have also been used for foot ulcer segmentation [18,19,20] and pressure ulcer segmentation and classification [21]. However, few studies have dealt with wound segmentation using DL techniques. Such studies do not provide the real-time segmentation of wound images and non-contact wound measurement without the help of gauges, but the proposed real-time Automatic Wound Segmentation Assessment (AWSA) framework addresses these concerns.

In this study, we performed the real-time automatic assessment of body surface wounds, which could help to rapidly assess wounds in many patients under emergency medical care and provide targeted treatment during the first 30 min of admission. This task mainly involved two steps: automatic wound segmentation and the assessment of the wound area. Wound image segmentation was conducted to locate the boundary between the wound and the surrounding skin [22]. The measurement of the wound area is usually performed manually, which is time-consuming and inaccurate and causes discomfort to patients [23,24,25]. Accurate and automatic wound measurement mostly relies on well-segmented wound regions. Previous studies lacked the required accurate segmentation, as they focused more on the retrieval and classification tasks.

We prepared the real-time AWSA framework to address the shortcomings of the presently used methods. Real-time AWSA uses CNNs and automatically detected and segmented wound areas delineated in images. It automatically calculates the area of the unknown wound by building a functional model between the prior graphics pixels and the graphics shooting heights. This method could help medical experts quickly obtain patients’ wound information without touching the wound or using measuring tools (such as measurement rulers or tapes). We used a novel and efficient short-term dense concatenate (STDC) network structure that removed structural redundancy to address these problems. Specifically, the basic module of the STDC network was formed by gradually reducing the dimensionality of feature maps and using their aggregation for image representation. In the decoder, we proposed a detail aggregation module by integrating the learning of spatial information into low-level layers in a single-stream manner. The low-level and deep features were fused to predict the final segmentation results. Finally, STDC-Net was used as the backbone to achieve a state-of-the-art speed–accuracy trade-off in real-time semantic segmentation by adding a coordinated attention mechanism.

The contributions of real-time AWSA are two-fold:The improved STDC-Net architecture with the pretrained weight model as the encoder layer achieved a trade-off between wound segmentation accuracy and prediction speed.The coordinated attention mechanism was proposed to better obtain the global receptive field and encode the accurate location information so that the network could locate the target of interest more accurately and further improve the performance of the proposed network.The wound area estimation without contact and without measurement tools was realized by constructing a functional relationship model between prior graphic pixels and image shooting heights.

This study involved an extensive experimental analysis comparing the proposed method with the currently used state-of-the-art segmentation methods, demonstrating the highly accurate real-time performance of real-time AWSA. Regarding wound area measurement, we compared the results obtained using the real-time AWSA method with manual segmentations based on methods used in previous studies, revealing the high-level accuracy of real-time AWSA.

Paper outline: The challenges facing surface wound area estimation and the basic concepts used in this study are discussed in Section 1. Section 2 discusses related studies. Section 3 details the real-time AWSA framework. Section 4 presents the materials and methods used in this study. Section 5 presents the experimental results and the corresponding discussion. Section 6 summarizes this study.

## 2. Related Works

To date, several studies have been conducted on wound segmentation and wound area estimation. For example, Chino et al. proposed Automatic Skin Ulcer Region Assessment (ASURA), a segmentation method for ulcer wounds based on U-Net [26]. However, some defects were found. First, the simple skip-connection between the encoder and the decoder did not account for the importance of different channels, which may have reduced the accuracy. Second, as the network became deeper and wider, the redundancy of the network structure made the segmentation task more time-consuming and difficult to optimize.Moreover, the ASURA system proposed by Chino et al. was used for the segmentation and measurement of wound area in real-world units (cm^2^) [27].

The SegNet decodes the feature map by upscaling and using a series of convolutions [28,29]. However, these networks require thousands of annotated training samples. Ronneberger et al. made a breakthrough in medical image segmentation using U-Net based on FCN to overcome this problem [30]. In U-Net, the decoder receives a copy of the output of the activation layers and concatenates it with the upscaling tensor. In this manner, U-Net can pass on the spatial information lost in the encoder step to the corresponding decoder layers, improving the segmentation output.

Many U-Net-based variants have emerged in recent years [31,32]. ASURA used U-Net as the network backbone to perform image segmentation and achieved decent accuracy. Meanwhile, ASURA automatically measured the wound size and adjusted the measurements manually through the app. However, this method was time-consuming for segmentation tasks due to redundant network structures and did not provide real-time performance when a large number of wound images needed to be processed in emergency medical care. Furthermore, measurement tools (measurement rulers/tapes) were required to estimate the wound area in the wound measurement task.

Dorileo et al. proposed an image segmentation method based on the analysis of the RGB channels of the image [33]. As all images had a blue background, they discarded the blue channel and used the intensity channel of the hue, saturation, and intensity (HSI) color space. For each channel, Dorileo et al.’s method helped automatically find thresholds and process the discovered regions by focusing on blobs near the center of the image. The main issue with this method was the need for a controlled environment.

Blanco et al. proposed QTDU, a deep-learning-based approach to analyze dermatological wounds using superpixels [34]. QTDU used CNNs for wound segmentation and relied on superpixel approaches to divide images into regions. However, QTDU did not segment the rules/tapes present in the images, and the estimation of the wound area involved counting the number of pixels inside the segmented area of each identified tissue and checking this value proportionally with the number of pixels of the entire image.

Seixas et al. employed off-the-shelf classifiers to segment wound images. They extracted pixel-wise color features, the mean value of the neighborhood of the pixel, and the difference in the pixel value and the mean beforehand. They segmented a training set of images to isolate the wound region.

Pereyra et al. proposed a segmentation process based on a multivariate Gaussian mixture model. The clusters were manually selected in a graphical user interface (GUI) to output the segmentation mask. Blanco et al. proposed the Counting-Labels Similarity Measure (CL-Measure), which focused on retrieving skin wound images based on visual similarity. Chino et al. proposed Imaging Content Analysis for the Retrieval of Ulcer Signatures (ICARUS), which was based on superpixels combined with Bags of Visual Signatures. It focused on the content-based retrieval of ulcer images and presented higher-quality results than CL-Measure.

Dastjerdi et al. proposed another method for semi-automatic wound segmentation and area measurement. It used both 2D and 3D representations, processing a single photo or a video, respectively. The 2D photo could be taken using a digital camera or smartphone, with a flexible paper ruler placed near the wound for size measurement. The segmentation started by roughly outlining the region of interest around the ulcer. Then, a trained random forest model calculated a probability map of each pixel belonging to the wound or healthy skin. A binary mask containing the wound area was created over the probability map by employing Otsu’s threshold. The ruler was segmented to calculate the ratio between pixels and centimeters. However, this segmentation method lacked real-time applications, and wound area estimation required the aid of a ruler.

CNNs are mainly used for image recognition tasks. They consist of a series of convolution operations that encode an image into a feature map and can be used to perform image segmentation. The fully convolutional network (FCN) is a pioneering work of CNN in image segmentation. 

DeepLab is another DL model for image segmentation. It employs dilated atrous convolutions to upscale the low-level features and enlarges the field of view of filters using a simple architecture. DeepLabv3+ is its latest version, which has an effective decoder to refine the segmentation, replacing the maximum pooling operations with depth-wise separable convolutions. The DeepLabv3+ decoder concatenates the encoded features, which are upscaled by a factor of four, with the corresponding features.

A comparison of our method with current state-of-the-art wound segmentation methods is presented in Table 1. Almost all methods could segment the wound. Some methods used superpixels in semantic segmentation to reduce the complexity of the image, which might impact the effect of wound segmentation. Real-time AWSA uses a novel and efficient STDC-Net classification network as a backbone to achieve high-precision wound segmentation with a high FPS while adding a coordinated attention mechanism to achieve the optimal speed–accuracy trade-off. However, none of the aforementioned methods realized the estimation of wound area without external measurement tools. This study measured unknown and irregular wounds by constructing the relationship model framework between prior graphics pixels and shooting heights, which could not only measure the wound area in real-world units but also realize the estimation of wound area without contact and without measurement tools.

## 3. Real-Time AWSA

We prepared a real-time wound segmentation and wound area estimation framework, real-time AWSA, that automatically measures wound area in images. Real-time AWSA uses deep CNNs to segment wounds. The functional relationship model between prior graphics pixels and image shooting heights was constructed to automatically measure the segmented wound area, which not only realized the estimation of wound area in real-world units but also avoided the use of measurement tools. Real-time AWSA works based on the following two main steps: (1) an automatic segmentation of surface wounds and (2) the construction of a functional relationship model between prior graphics pixels and graphics shooting heights to automatically estimate the wound area. Figure 1 shows the real-time AWSA framework. Real-time AWSA also offers an interactive GUI in which the user can obtain the predicted segmentation mask for the wound and interact with the GUI to obtain the estimated wound area.

### 3.1. Real-Time Wound Segmentation

In the segmentation task, real-time AWSA received RGB images of the surface wound and output segmentation masks with the wounds. The wound segmentation process was based on a convolutional deep neural network developed for image segmentation. Due to the limited training dataset and considering the tradeoff between segmentation accuracy and speed, real-time AWSA used the STDC-Net architecture, which could address the issues of the possible trade-off between segmentation accuracy and speed [35].

Figure 2 shows the improved network architecture in this study. The network consisted of an encoder and a decoder. First, real-time AWSA received, as input, an RGB image with an arbitrary resolution. As the input layer of the network was a tensor of size 512 × 512 × 3, the image was resized to a 512 × 512 resolution. The network architecture consisted of six stages, in addition to an input layer and prediction layer. Generally, stages 1–5 down-sampled the spatial resolution of the input with a stride of two, and stage 6 output the prediction logits by one ConvX, one global average pooling layer, and two fully connected layers. Each ConvX consisted of one convolutional layer, one batch normalization layer, and one ReLU activation layer. Stages 1 and 2 are usually regarded as low-level layers for appearance feature extraction. We used only one convolutional block each in stages 1 and 2, which proved to be effective. The number of STDC modules in stages 3 to 5 was carefully tuned in our network. The first STDC module in each of these three stages down-sampled the spatial resolution with a stride of two. The following STDC modules in each stage kept the spatial resolution unchanged. We used the attention refine module to refine the combination features of stages 3 to 5. We adopted the feature fusion module in BiSeNet for the final semantic segmentation prediction [28] to fuse the 1/8 down-sampled feature from stage 3 in the encoder and the counterpart from the decoder. We set the output channel number as 1024 and carefully tuned the channel number of the remaining stages until reaching a good trade-off between accuracy and efficiency. Further, a coordinated attention module was added before and after the feature fusion module, which further improved the network prediction ability. Finally, the output tensor was resized to the resolution of the input image.

### 3.2. Wound Area Estimation

After wound segmentation, real-time AWSA estimated the wound area in real-world units by constructing a functional relationship model between prior graphic pixels and the shooting heights of the image. Figure 3 shows the steps of real-time AWSA for estimating the wound area ***S_w_***. Algorithm 1 shows how real-time AWSA estimated the wound area ***S_w_***.
**Algorithm 1** Wound area estimation.**Initialization**: ***S_pi_*** = area of prior graphics**Require: *I***: *input image,*
**Mask***: segmentation mask***Output: *S_w_*** = *the area of the wound*Begin1. **λ*_i_***: obtained prior graphics pixels (**I**,**Mask**)2. λ_1_, λ_2_, …, λ*_n_*: pixels of different shooting heights *h*_1_, *h*_2_, …, *h_n_*3. λ = *f* (*h*): polynomial fitting of shooting height and prior graphics pixels4. **if** (*h_i_* < *h*_1_) or (*h_i_* > *h_n_*):   **return none**5. **elif** *h*_1_ < *h_i_* < *h_n_*:6.   ***S_w_*** = φ7.   **for** *h_i_* **in range**(*h*_1_, *h_n_*) **do**8.      **λ*_w_*** = wound image pixels obtained9.      calculate the wound area ***S_w_*** = (***S_pi_*** × **λ*_w_***)/**λ*_i_***10.   **end for**11. **end if**12. **return *S_w_***

Prior graphics took regular shapes such as triangles or squares with known areas. A smartphone was used to carry a laser ranging sensor to photograph prior graphics whose area *S_pi_* was known. The shooting height varied from the minimum height *h*_1_ to the maximum height *h_n_*, and the distance was equally divided, as shown in Figure 3. The images were taken using the same smartphone with the same resolution. The number of pixels occupied by the prior graphics in the image changed with the shooting height, and each shooting height *h* corresponded to a pixel number *λ*. Next, the discrete relationship between the image shooting heights *h*_1_, *h*_2_, *h*_3_, …, *h_n_* and the number of pixels occupied by the 2D prior graphics *λ*_1_, *λ*_2_, *λ*_3_, …, *λ_i_* was constructed using a polynomial fitting method, i.e., *λ* = *f* (*h*). From this function, we could determine the number of pixels *λ_i_* of the image corresponding to *h_i_* at any point in the range *h*_1_ to *h_n_*. Last, we took a wound image with an unknown area *S_w_* at a height greater than *h*_1_ and lower than *h_n_*, and then the wound area *S_w_* could be obtained as follows:(1)$$S_{w}=\frac{S_{pi}}{\lambda_{i}}\times \lambda_{w}$$
where *λ_w_* is the number of pixels in the image occupied by the wound at the shooting height *h_i_*.

### 3.3. Graphical User Interface (GUI)

The real-time AWSA framework contains an interactive GUI that allows users to view the original wound image, the ground truth of the wound, and the predicted MASK of the wound after image segmentation. The interactive interface also allows the user to obtain the segmented wound area in real time after obtaining the predicted mask of the wound. Figure 4 shows the GUI of the real-time AWSA framework. The input wound image is below the Original Image heading. The ground truth of the wound is under the Label Image heading, while the prediction mask after wound segmentation is under the Prediction Image heading. The function selection of the interactive interface is on the left side of the GUI. Users can select any wound image and load the model to complete the semantic segmentation of the wound region. Real-time AWSA enables automatic wound area estimation. The user enters the height at which the wound image was taken, and real-time AWSA measures the area of the segmented wound in real-world units.

## 4. Materials and Methods

The performance of real-time AWSA was evaluated to verify its rapid wound area estimation. Two sets of experiments were conducted for performance evaluation: real-time wound segmentation and wound area estimation. All experiments were conducted using a 4.20-GHz Intel Core i7-12700F CPU, 32 GB RAM, and 12 GB NVIDIA GTX 3060Ti GPU. Further, we implemented real-time AWSA in Python based on the PyTorch1.8.1 framework, and the development software used was Pycharm 3.7.

### 4.1. Datasets and Pre-Processing

Real-time AWSA was evaluated on the self-built wound dataset WOUND. It consisted of 661 images of different wound types, including scratches, cuts, and bruises, mainly on the arms, legs, and upper body. Among them, 535 images were used as the training set, and 126 images were used as the test set. In the images, the wounds were located all over the body, and some patients had multiple wounds. The image dataset WOUND was obtained from the National Trauma Database and Chengde County Hospital. All images in the dataset were captured by digital cameras. For the dataset WOUND, the experts manually segmented the wound area to create the ground truth mask. Figure 5 shows several wound images and their respective ground truth masks, where the gray area is the wound. We augmented the dataset and compared our method with currently used state-of-the-art wound image segmentation methods to evaluate the performance of real-time AWSA.

DL models require a large amount of data for training to improve [10]. As WOUND was a small dataset, a data augmentation technique was used to improve the robustness of real-time AWSA. A series of methods, such as random flipping, random cropping, Gaussian noise, and adjusting brightness, were applied to enhance the number of images and masks. The flip angle of the image was randomly selected between 0° and 360°, and the image was randomly cropped to one third of the height or width. Figure 6 shows the results of image data augmentation; each transformed image had its corresponding mask. Table 2 depicts the quantitative comparison of original and enhanced images.

### 4.2. Wound Area Estimation

In this section, we detail the ability of real-time AWSA to estimate wound area in real-world units. Real-time AWSA was also evaluated in regard to its ability to estimate wound area in real-world units using the scale relationship between prior graphics with a known area and a wound with an unknown area, without considering pixel density. The number of pixels in the image was calculated using a computer, and the prior graphics were taken using the same smartphone as for the wound, with no resolution difference. The wound area in real-world units was calculated using Equation (1):Real area (*S_wReal_*): the area of the ground truth mask.Estimated area (S*_wEst_*): the estimated area of the wound region.MBR area (*S_wMbr_*): wound area determined using a manual measurement method.

To evaluate the ability of real-time AWSA to estimate the wound area, we used Equation (2) to calculate the percentage error *E* for evaluating the ability of real-time AWSA to estimate wound area, where ***s*** is the true area and $\hat{s}$ is the estimated area.
(2)$$E=\frac{\left|s-\hat{s}\right|}{S}\times 100\%$$

### 4.3. Experimental Details

A gradient descent decay operation was used to find the learning rate during the training. The initial learning rate was set to 0.01, and the batch size during training was set to 8. Furthermore, we employed model training using pretrained weights and compared it with training from scratch, as shown in Figure 7. In terms of validation loss, training with pretrained weights converged to 0.015 after around 20 epochs, while training from scratch fluctuated around 0.04 even at the end of training. From epoch 3, the validation dice value of the model with pretrained weights showed better performance than the training from scratch. As depicted in the figure, the validation dice value of the pretrained weight strategy was stable at around 0.995, while the training strategy from scratch was stable at around 0.981.

## 5. Results and Discussion

The performance of real-time AWSA in terms of segmenting wound images and estimating wound area is detailed in this section.

### 5.1. Testing Metrics

The calculation formula for wound segmentation accuracy is as follows:(3)$$mAP=\frac{1}{\left|Q_{R}\right|}\sum_{q\in Q_{R}}AP(q)$$

*mAP* is an important indicator to measure the accuracy of segmentation, where *Q_R_* represents the number of verification datasets.

*m-IoU* is the average of the intersection-over-union ratio of the real label and the predicted segmentation. The larger the ratio, the more accurate the segmentation. The formula is as follows:(4)$$mIoU=\frac{1}{k+1}\sum_{i=0}^{k}\frac{p_{ii}}{\sum_{j=0}^{k}p_{ij}+\sum_{j=0}^{k}p_{ji}+p_{ii}}$$
where *p_ij_* represents predicting category *i* as category *j*.

The recall represents the ratio of the predicted wound area to the real wound area. The closer the ratio is to 1, the more accurate the segmentation. The formula is as follows:Recall = TP/(TP + FN)(5)

The dice score is the harmonic mean of precision and recall and reflects the segmentation accuracy.

### 5.2. Segmentation Performance of Different Network Structures on Two Wound Datasets

We adopted the STDC-Net classification network as the backbone of the segmentation model in this study. The models were divided into STDC-Net813 and STDC-Net1446 based on the complexity of the model. Experiments were conducted on two wound datasets, WOUND-1 and WOUND-2, to demonstrate the effectiveness of our adapted model. Considering STDC-Net as the benchmark, the models were divided into eight categories based on whether they used pretrained weights or added a coordinated attention mechanism. Each segmentation method was implemented on each image in each test dataset to evaluate the effectiveness of all models. Then, we calculated six metrics: mAP, mIoU, recall, dice score, FPS, and AUC. Table 3 presents the results obtained for all networks using WOUND-1 and WOUND-2. As depicted in the table, the model using pretrained weights had better segmentation accuracy, and the network segmentation accuracy was further improved using the coordinated attention mechanism. All indicators were improved, but the FPS was reduced by about 9%. Further, the performance of STDCNet_CA813_Pretrain was slightly lower than that of STDCNet_CA1446_Pretrain in terms of dice score and AUC. On comparing the FPS, that of STDCNet_CA813_Pretrain was found to be 25% higher than that of STDCNet_CA1446_Pretrain, reflecting the higher processing speed of the network. Because of the complexity of the network, STDCNet_CA813_Pretrain was slightly inferior to STDCNet813 in terms of processing speed, but was significantly better than STDCNet813 in terms of segmentation accuracy. Therefore, considering all aspects, we selected STDCNet_CA813_Pretrain as the model for real-time AWSA.

### 5.3. Comparison of Real-Time AWSA with State-of-the-Art Methods

Next, we compared real-time AWSA with state-of-the-art models, mainly including ASURA (U-Net) and DeepLabv3+. Chino et al. proved that ASURA using U-Net as its backbone outperformed CL-Measure, superpixel-based ICARUS, and DL-based QTDU. Therefore, we mainly used ASURA with U-Net as the backbone and the general image semantic segmentation model DeepLabv3+ as the comparison object. As depicted in Table 4, our model performed best on the mAP, m-IoU, recall, dice score, FPS, and AUC metrics. Specifically, the mAP, m-IoU, recall, and dice score values of real-time AWSA on the two wound datasets were about 0.14%, 2.7%, 0.44%, and 0.14% higher than those of the second-ranked method, ASURA (U-Net), respectively. This confirmed that our improved model had a better segmentation capability. Meanwhile, the FPS values of our method for the two wound datasets were 100.08 and 102.11, which were about 42% higher than those of the second-ranked method, Deeplabv3+, reflecting better real-time performance. Further, the AUC best reflected the overall segmentation performance of the models. In this study, our network achieved AUC levels of 0.9938 and 0.9949, demonstrating the robustness of our method.

Figure 8 presents the segmentation outputs from the WOUND-1 and WOUND-2 datasets. Using both datasets, the output of real-time AWSA was extremely close to the ground truth. Due to the relatively small size of the training datasets, Deeplabv3+ faced issues in correctly segmenting wounds. When we compared the detailed visual results of ASURA with U-Net as its backbone and the findings of our method, as shown in Figure 8, we found that it achieved significant results but still lacked enough semantic information and had obvious false-positive segmentation. These findings proved that our method was more robust.

As the AUC value, which is essentially the area under the receiver operating characteristic (ROC) curve, can comprehensively reflect the segmentation capability, we compared the ROC curves of the different models. Figure 9 shows that our network outperformed the others for both datasets.

### 5.4. Wound Area Estimation

We evaluated the ability of real-time AWSA to measure wound area in real-world units, such as cm^2^. We calculated the wound areas in the test dataset that had already been measured by medical experts at the hospital, and these measured wounds served as standards. Meanwhile, we calculated the error of the manual measurement method and the method proposed in this study relative to the expert standard. The results presented in this section are the average of all test images.

Figure 10 shows the automatic wound area estimation system, consisting of a height platform, a smart phone, a ranging laser sensor, and a personal computer. First, a smartphone was used to carry laser ranging sensors to collect images of prior graphics at different heights. In this study, the shooting height ranged from 100 to 500 mm, with an interval of 10 mm. According to the collected image data, the polynomial fitting method was used to build the relationship model between prior graphics pixels and shooting heights (Figure 8). In the next step, the wound images were captured at any height from 100 mm to 500 mm using the same method. The same camera was used to acquire prior images and wound images with the same number of pixels. Finally, the segmented wound area was calculated using Equation (1).

Figure 11 shows some examples of area estimation. We measured the area of the wound images from two test datasets using three area estimation methods and obtained the relative error against the gold standard of human experts. As depicted in Table 5, the relative errors of the area estimates using the MBR method for the two datasets were 30.5% and 33.9%, respectively, and the errors using the thin-film edge labeling method were 8.1% and 6.3%, respectively. Although the thin-film edge labeling method demonstrated very small errors, the wound was not of a regular shape, and the wound edge might not have covered a full square grid, reducing the accuracy of the area estimation. Moreover, the counting of squares is time-consuming and laborious, delaying wound treatment. Notably, the errors in wound area estimation using the method proposed in this study were 3.7% and 3.1%, showing the best estimation results. Importantly, this method did not rely on measurement tools and exhibited better real-time performance.

## 6. Conclusions

In this study, we explored methods for evaluating large-scale wounds in emergency situations and proposed the real-time AWSA framework to automatically segment wound images and estimate the area of a wound. Real-time AWSA used the STDC-Net classification network as its backbone, eliminated structural redundancy, adopted a pretrained weight model, and improved the network processing speed while accurately segmenting the wound, realizing a segmentation accuracy–prediction speed trade-off. A coordinated attention mechanism was introduced to further improve the network segmentation performance. Moreover, we constructed a functional relationship model between prior graphics pixels and shooting heights to perform wound area measurements without contact and measurement tools. We evaluated real-time AWSA using two wound datasets, WOUND-1 and WOUND-2, and found that the accuracy was greatly improved compared with the current state-of-the-art methods. The experimental results showed that real-time AWSA outperformed the state-of-the-art methods in terms of mAP, mIoU, recall, and dice score. The AUC value, which most reflected the comprehensive segmentation capability, also reached the highest level of about 99.5%. The FPS values of our proposed segmentation method in the two wound datasets were 100.08 and 102.11, respectively, which were about 42% higher than those of the second-ranked method, reflecting better real-time performance. Further, real-time AWSA could automatically estimate the wound area in square centimeters with relative errors of only 3.7% and 3.1% in the two test datasets, respectively, showing the best estimation results.

The method proposed in this study could quickly process a large number of collected wound images for trauma treatment in emergency environments, areas with scarce medical resources, or trauma patients with limited mobility. The main tasks included the automatic segmentation of the wound area and automatic estimation. The wound information could be uploaded and sent to telemedicine experts to achieve immediate treatment and real-time wound care. The current disadvantage is that the method in this paper was mainly aimed at determining the two-dimensional area of a wound, without considering the curvature factor and depth information. In addition, the deep learning method may lose some semantic information in areas where the color transition of the wound is not clear. In the future, deep learning combined with 3D reconstruction could deal with more complex wounds and solve these problems, which is our current research focus. 

## Figures and Tables

**Figure 1 healthcare-11-01205-f001:**
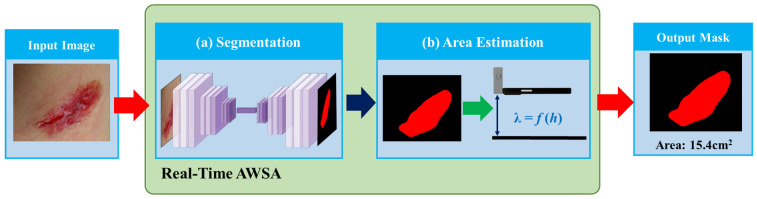
Real-time AWSA framework. (**a**) Image segmentation network. (**b**) Wound area estimation method.

**Figure 2 healthcare-11-01205-f002:**
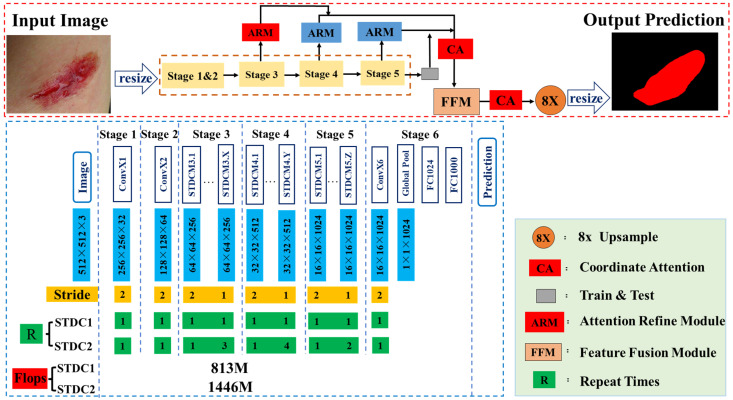
Overview of the real-time AWSA network. ARM denotes attention refine module, and FFM denotes feature fusion module.

**Figure 3 healthcare-11-01205-f003:**
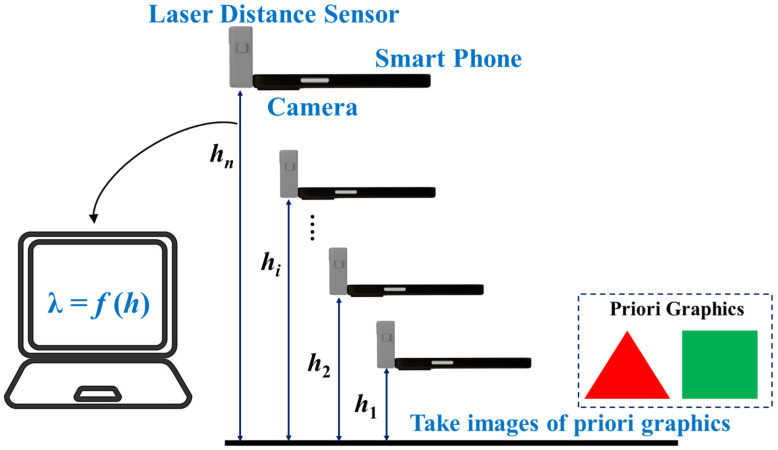
Functional relationship model between prior graphics pixels and shooting heights to estimate the wound area in real-world units.

**Figure 4 healthcare-11-01205-f004:**
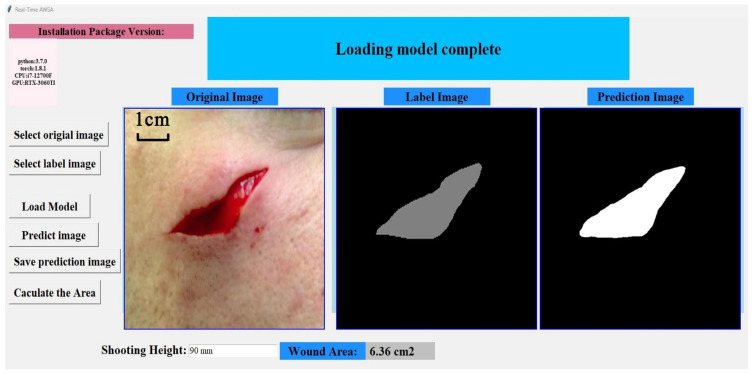
Real-time AWSA interactive graphical user interface.

**Figure 5 healthcare-11-01205-f005:**
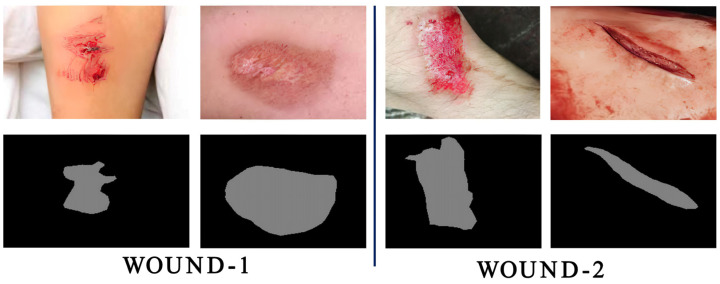
Example wound images. The dataset images are on the top row, and the ground truth masks are on the bottom row.

**Figure 6 healthcare-11-01205-f006:**
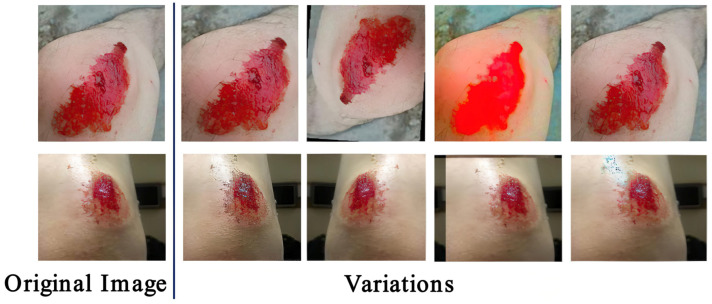
Examples of images generated by data augmentation.

**Figure 7 healthcare-11-01205-f007:**
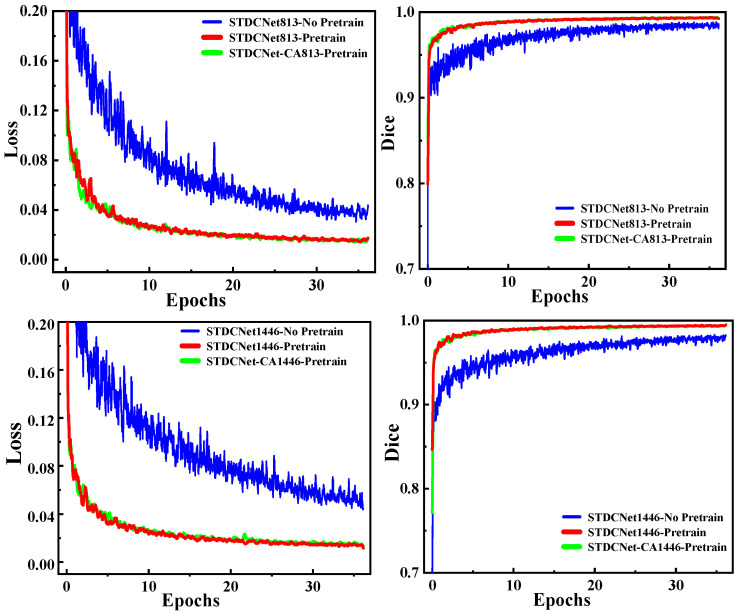
Comparison of training using pretrained weights versus training from scratch along with training epochs. (**Left**): comparison of validation loss between two training strategies. (**Right**): comparison of validation dice between two training strategies.

**Figure 8 healthcare-11-01205-f008:**
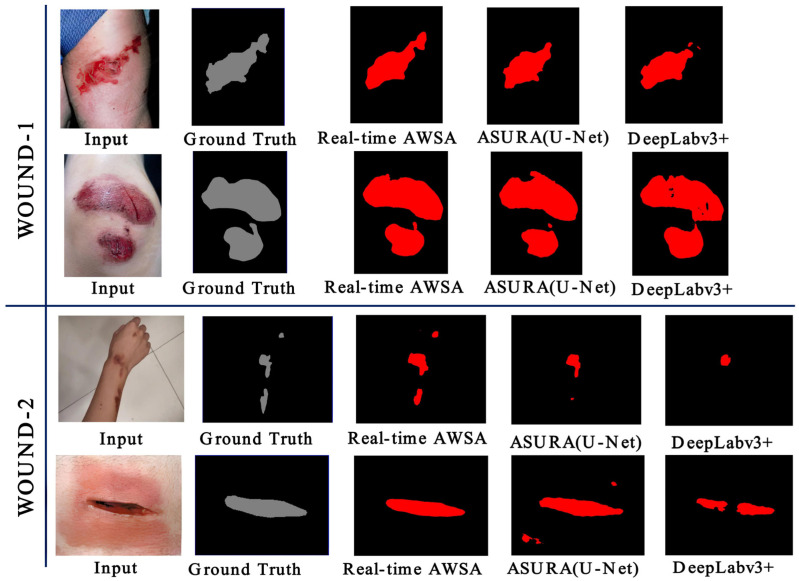
Wound segmentation of images from WOUND-1 and WOUND-2.

**Figure 9 healthcare-11-01205-f009:**
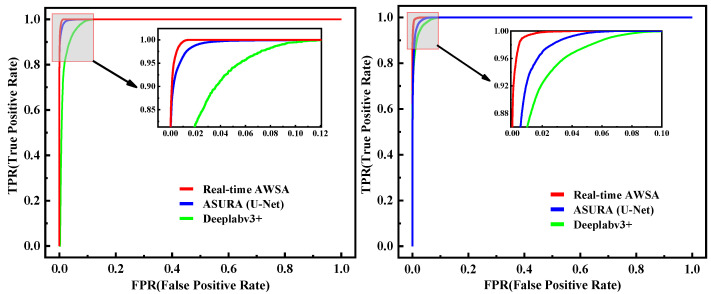
ROC curves of different models for wound segmentation. (**Left**): WOUND-1, (**Right**): WOUND-2.

**Figure 10 healthcare-11-01205-f010:**
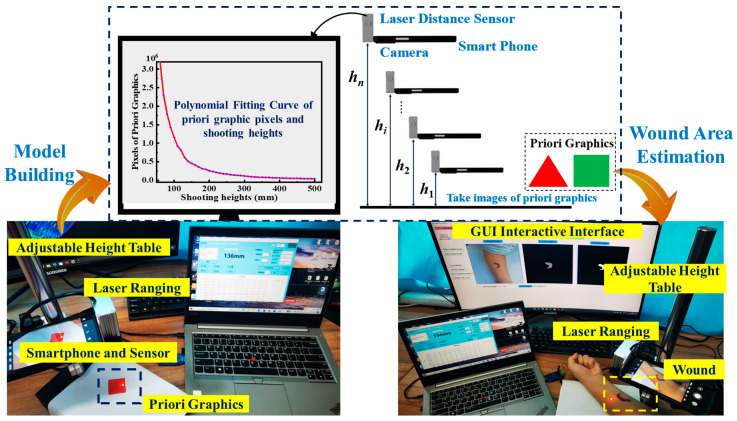
Automatic wound area estimation system, consisting of a height platform, a smart phone, a ranging laser sensor, and a personal computer.

**Figure 11 healthcare-11-01205-f011:**
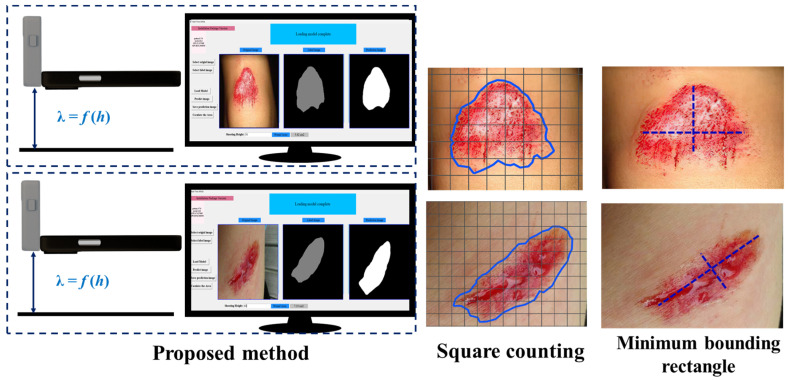
Area estimation for two wound datasets using the real-time AWSA framework.

**Table 1 healthcare-11-01205-t001:** Summary of different methods to segment wounds.

	Wound Segmentation	Real-Time Segmentation	Detect Measurement Tool	Area Assessment without Measuring Tools
Dorlieo	√			
Seixas	√			
Pereyra	√			
CL-Measure	√			
ICARUS	√			
Dastjerdi	√			
ASURA (U-Net)	√		√	
Real-Time AWSA	√	√	√	√

**Table 2 healthcare-11-01205-t002:** Number of images in each dataset.

Training Dataset	Size	Size after Augmentation
WOUND-1	274	1826
WOUND-2	261	1722

**Table 3 healthcare-11-01205-t003:** Evaluation of the segmentation methods for each dataset. The values marked with * are the best results. All values are percentages.

**WOUND-1**						
**Model**	**mAP**	**m-IoU**	**Recall**	**Dice Score**	**FPS**	**AUC**
STDCNet813	0.8981	0.8274	0.9766	0.7541	111.78 *	0.9813
STDCNet813_Pretrain	0.9321	0.8736	0.9837	0.8331	110.66	0.9932
STDCNet_CA813	0.9062	0.8444	0.9866	0.7621	100.12	0.9841
Real-Time AWSA (STDCNet_CA813_Pretrain)	0.9481 *	0.8928 *	0.9877 *	0.8473	101.08	0.9938
STDCNet1446	0.8749	0.8011	0.9695	0.6831	84.26	0.9779
STDCNet1446_Pretrain	0.9333	0.8713	0.9824	0.8401	85.32	0.9934
STDCNet_CA1446	0.8904	0.8026	0.9672	0.7047	78.02	0.9784
STDCNet_CA1446_Pretrain	0.9462	0.8852	0.9831	0.8496 *	76.49	0.9951 *
**WOUND-2**						
**Model**	**mAP**	**m-IoU**	**Recall**	**Dice Score**	**FPS**	**AUC**
STDCNet813	0.8935	0.8441	0.9773	0.7632	111.33 *	0.9783
STDCNet813_Pretrain	0.9377	0.8739	0.9896	0.8347	109.38	0.9938
STDCNet_CA813	0.8995	0.8427	0.9881	0.7597	101.58	0.9841
Real-Time AWSA (STDCNet_CA813_Pretrain)	0.9477 *	0.8944 *	0.9883 *	0.8495	102.11	0.9949
STDCNet1446	0.8767	0.8019	0.9672	0.6822	82.27	0.9822
STDCNet1446_Pretrain	0.9339	0.8762	0.9864	0.8451	85.88	0.9954
STDCNet_CA1446	0.8924	0.8077	0.9692	0.7067	78.59	0.9761
STDCNet_CA1446_Pretrain	0.9471	0.8905	0.9882	0.8511 *	77.24	0.9952 *

**Table 4 healthcare-11-01205-t004:** Results of different algorithms for WOUND-1 and WOUND-2 datasets. The bold values are our proposed model, and the values marked with * are the best results. All values are percentages.

**WOUND-1**						
**Model**	**mAP**	**m-IoU**	**Recall**	**Dice Score**	**FPS**	**AUC**
Deeplabv3+	0.8931	0.8293	0.9757	0.7604	58.45	0.9811
ASURA(U-Net)	0.9319	0.8690	0.9833	0.8463	55.04	0.9889
**Real-Time AWSA**	**0.9451 ***	**0.8928 ***	**0.9877 ***	**0.8473 ***	**100.08 ***	**0.9938 ***
**WOUND-2**						
**Model**	**mAP**	**m-IoU**	**Recall**	**Dice Score**	**FPS**	**AUC**
Deeplabv3+	0.8947	0.8255	0.9713	0.7604	58.74	0.9788
ASURA(U-Net)	0.9332	0.8697	0.9839	0.8471	54.22	0.9898
**Real-Time AWSA**	**0.9477 ***	**0.8944 ***	**0.9883 ***	**0.8485 ***	**102.11 ***	**0.9949 ***

**Table 5 healthcare-11-01205-t005:** Relative error (%) of the three wound area estimation methods compared to the human expert gold standard.

Datasets	Relative Error (%)			
	Number	Proposed Method	Square Counting	MBR Method
WOUND-1	30	3.1	6.3	30.5
WOUND-2	30	3.7	8.1	33.9

## Data Availability

No new data were created.

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
