# Peer review of "Convolution Neural Network with Coordinate Attention for Real-Time Wound Segmentation and Automatic Wound Assessment"

_healthcare, 2023, doi:10.3390/healthcare11091205_

Round 1
Reviewer 1 Report
COMMENTS TO THE AUTHOR(S)
The authors propose an exciting study on the “Convolution Neural Network with Coordinate Attention for Real-Time Wound Segmentation and Automatic Wound Assessment”. However, there are some issues with the current study that need to be included and/or clarified before the publication, as detailed below:
1. What are FPS, AUC, mAP and many others? Provide the full names at the first occurrence. In later text, abbreviated forms should be used.
2. References are missing in many statements in the introduction section author should reference the current literature.
3. The quality of the images is very poor. Therefore, it is suggested that authors must enhance the text quality and font size in the figures to visualize clearly.
4. Authors must compare their results with other techniques and research studies in the discussion section.
5. What is the main novelty in your proposed technique than other and what are the prospects of this study?
6. Which types of problems were faced while processing different wounds with this proposed technique?
7. The authors used the old literature for reference. Therefore, the authors should use more recent research (2021 to 2023) to reference this study.
8. In the references section, the authors used different formats; some information is missing. Authors should follow proposed formats for references consistently throughout the manuscript.
9. The manuscript should be checked thoroughly for grammatical and typing mistakes, as there are some mistakes throughout the manuscript.
Author Response
- Respond to Reviewer 1:
(1) The full name of FPS is Frame Per Second, which represents the speed of wound segmentation of the improved convolutional neural network proposed in this paper.
The full name of AUC is Area Under Curve, which is an evaluation index of semantic segmentation model. AUC is defined as the area under the ROC curve. The ROC curve takes the true positive rate (TPR) as the vertical axis and the false positive rate (FPR) as the horizontal axis. The ROC curve ranges from coordinates (0,0) to coordinates (1,1), the closer the curve is to the upper left corner, that is, the closer the AUC is to 1, the better the segmentation performance of the model.
The full name of mAP is Mean Average Precision, which indicates the index of semantic segmentation accuracy.
The full name of mIoU is mean Intersection over Union. The Intersection over Union represents the correlation between the real value and the predicted value, and is an evaluation index for semantic segmentation.
According to the reviewer’s comments, we provided the full name of the abbreviation when it first appeared, and continued to use the abbreviation later in the article.
(2) According to the reviewer’s comments, the references have been revised.
(3) According to the comments of the reviewer, we have improved the quality of the image and increased the resolution of the image to make the text in the figure clearer.
(4) According to the reviewers' comments, we compared the method proposed in this paper with the two most popular and effective wound segmentation methods in section "7.3 Comparison of Real-Time AWSA with the state-of-the-art methods", Especially the ASURA framework proposed by Chino et al. ASURA has been proven to be better than other wound semantic segmentation methods, so this paper compares with ASURA to prove the superiority of this study. In the two wound datasets, six evaluation indexes of image semantic segmentation are compared.
(5) The main purpose of this article is to explore a method for rapid assessment of wound size in an emergency setting. In the case of battlefields, accidents and disasters, where there are many wounded, rapid wound assessment can achieve immediate treatment without delaying the injury. Therefore, while showing the accuracy of wound segmentation, this study also needs to reflect the real-time segmentation, that is, faster reasoning speed. This research has good application prospects, especially in the current background of frequent natural disasters and safety accidents all over the world.
(6) In dealing with different wounds, two problems were encountered. The first one is that the color transition of some wounds is not obvious and the boundary is not clear. Our method sometimes loses some semantic information, resulting in low accuracy of wound segmentation. The second is that some wounds are infected, and the granulated information caused by these infections may be lost during image segmentation.
(7) According to the reviewer’s comments, the references have been revised, adding more recent research (2021 to 2023) to reference this study.
(8) The formatting of the references has been revised based on the reviewers' comments.
(9) According to the reviewer’s comments, the grammar and writing of the full text were checked and revised.
Please see the attachment.

Reviewer 2 Report
Some improvements must be made before it can be considered for publication. My comments are:
1. I recommend improving the following Figures because the resolution is not good and the text is blurred: Fig. 1, 2, 4, 8, and 10.
2. Figure 3 is missing from the text.
3. Citation number [25] is missing.
4. Please pay special attention to the references throughout the work, over 70% do not coincide with the text. For example, no reference matches the description from line 126 to line 200.
5. There are spelling mistakes in the text and in the tables, please correct them. For example, rows: 132, 227, 290, 414, 415, etc.
Author Response
- Respond to Reviewer 2:
(1) According to the comments of the reviewer, we have improved the quality of the image and increased the resolution of the image to make the text in the figure clearer.
(2) According to the comments of the reviewer, Figure 3 has been added.
(3) According to the comments of the reviewer, we added the citation [25].
(4) According to the comments of the reviewer. Previously, it was due to the loss of a reference, which caused a mismatch with the main text, and it has been revised now.
(5) According to the reviewer’s comments, corrections have been made to spelling errors in the text and tables.
Please see the attachment.

Reviewer 3 Report
Review comments for Convolution Neural Network with coordinated attention for real-time wound segmentation and automatic wound assessment.
We thank the author for investigating and evaluating the real-time automatic wound segmentation assessment measurement model for better wound size detection. This research paper is informative but not novel enough. Every step and description are organized and clear. Some minor concerns need to be addressed first before publication.
1. It would be beneficial to add scale bars to the digitally analyzed images.
2. Could this wound segmentation assessment detect all deep the wound is?
3. More experimental data should be added. Would it be possible to have this assess different types of wounds? such as diabetic foot ulcer, or surgical trauma?
Author Response
- Respond to Reviewer 3:
(1) Scale bars have been added to Figure 4 following reviewer’s comments.
(2) This research mainly focuses on the 2D semantic segmentation of wounds, and does not involve the estimation of 3D size. The depth information requires 3D reconstruction of the wound, and the meshing is used to calculate the depth and volume information. The 3D reconstruction of wounds is what I am currently researching. But this article does not cover these contents.
(3) Dear reviewers, diabetic foot ulcers and surgical wounds need in-depth information to better guide treatment. This paper is mainly aimed at the evaluation of scratches, cuts and bruises in the emergency environment.
Please see the attachment.

Reviewer 4 Report
Thorough and articulate article.
Great potential from this development, especially in emergency areas or remote regions; showed great improvement on current strategies.
Would this technology be able to calculate areas in wounds with bacterial presence?
Minor typo in Table 1.
The text in the figures is difficult to read, can higher quality images be used instead.
Author Response
(1) Our method is able to segment and calculate the bacterially infected portion of the wound if the bacterially infected portion is distinct compared to the unwounded skin. But if the color transition is not very clear, some semantic information may be lost during the segmentation process.
(2) According to the reviewer’s comments, errors in Table 1 have been corrected.
(3) According to the comments of the reviewer, we have improved the quality of the image and increased the resolution of the image to make the text in the figure clearer.
Please see the attachment

Reviewer 5 Report
Dear authors,
the following aspect should be edited in the manuscript "Convolution Neural Network with Coordinate Attention for Real-Time Wound Segmentation and Automatic Wound Assessment".
Abstract:
Please shorten the abstract and obtain a clear structure with "background, methods, results, conclusion". Please simplify the method section.
Introduction:
The introduction is about 9 pages long. Through the numerical order, it is for the reader not clear, if e.g. "2 Wound Area Measurement" is a part of the "1 Introduction" or is something separate. Please cut sharply the whole introduction with the focus on the aim of this study. At the end of the introduction, the study aim should be addressed / introduced.
Methods:
If available, please report the distribution of the included wound types of the "WOUND dataset".
Results / Discussion:
A clear distinction in this section between "results" and "discussion" is partly not possible. The result and discussion sections should be written separately. Please enhance the discussion in regard to clinical advantages / disadvantages and clinical future perspective.
The limitation Section is missing.
In almost all figures/tables abbreviations explantations are missing, please add them.
Author Response
(1) According to the reviewer’s comments, Abstracts have been shortened and presented in a clear structure, and the Methods section in the abstract has been simplified.
(2) According to the reviewer’s comments, the introduction section has been revised.
(3) According to the reviewer’s comments, we report the wound types in the wound dataset. (It consisted of 661 images of different types of wounds, including scratches, cuts and bruises, mainly on the arms, legs and upper body. Among them, 535 images were used as training set and 126 images were used as test set.)
(4) Dear reviewer, I would like to explain to you that in the same type of articles, the results and discussions are put together. Because this article is divided into three parts for results and discussion, if they are separated, the structure of the article will be unclear.
According to the reviewer’s comments, I have added clinical advantages/disadvantages as well as a discussion of future prospects. I also put the following text in the conclusion of the article.
The method proposed in this study can quickly process a large number of collected wound images for trauma treatment in emergency environments, areas with scarce medical resources, or trauma patients with limited mobility. The main tasks include automatic segmentation of wound area and automatic estimation. Upload wound information to telemedicine experts to achieve immediate treatment and real-time care of wound. The current disadvantage is that the method in this paper is mainly aimed at the two-dimensional area of the wound, without considering the curvature factor and depth information. In addition, the deep learning method may lose some semantic information in areas where the color transition of the wound is not clear. In the future, deep learning combined with 3D reconstruction will be able to deal with more complex wounds and solve these problems, which is what we are doing now.
Please see the attachment.

Round 2
Reviewer 1 Report
The authors completed all the necessary corrections in the revised manuscript and successfully attempted all queries. Now, this article can be accepted in its present form.
Reviewer 2 Report
The article can be accepted